# Unsupervised Learning of View-invariant Action Representations

**Junnan Li**
Grad. School for Integrative Sciences and Engineering
National University of Singapore
Singapore
lijunnan@u.nus.edu

**Yongkang Wong**
School of Computing
National University of Singapore
Singapore
yongkang.wong@nus.edu.sg

**Qi Zhao**
Dept. of Computer Science and Engineering
University of Minnesota
Minneapolis, USA
qzhao@cs.umn.edu

**Mohan S. Kankanhalli**
School of Computing
National University of Singapore
Singapore
mohan@comp.nus.edu.sg

## Abstract

The recent success in human action recognition with deep learning methods mostly adopt the supervised learning paradigm, which requires significant amount of manually labeled data to achieve good performance. However, label collection is an expensive and time-consuming process. In this work, we propose an unsupervised learning framework, which exploits unlabeled data to learn video representations. Different from previous works in video representation learning, our unsupervised learning task is to predict 3D motion in multiple target views using video representation from a source view. By learning to extrapolate cross-view motions, the representation can capture view-invariant motion dynamics which is discriminative for the action. In addition, we propose a view-adversarial training method to enhance learning of view-invariant features. We demonstrate the effectiveness of the learned representations for action recognition on multiple datasets.

## 1 Introduction

Recognizing human action in videos is a long-standing research problem in computer vision. Over the past years, Convolutional Neural Networks (CNNs) and Recurrent Neural Networks (RNNs) have emerged as the state-of-the-art learning framework for action recognition [3, 6, 35, 51]. However, the success of existing supervised learning methods is primarily driven by significant amount of manually labeled data, which is expensive and time-consuming to collect.

To tackle this problem, a stream of unsupervised methods have recently been proposed [8, 23, 33, 34, 53], which leverage *free* unlabeled data for representation learning. The key idea is to design a surrogate task that exploits the inherent structure of raw videos, and formulate a loss function to train the network. Some works design the surrogate task as constructing future frames [53] or future motion [33], while others use the temporal order of video frames to learn representations in a self-supervised manner [8, 23, 34]. Although they show promising results, the learned representations are often view-specific, which makes them less robust to view changes.

Generally, human action can be observed from multiple views, where the same action appears quite different. Therefore, it is important to learn discriminative view-invariant features, especially for action recognition from unknown or unseen views. Humans have the ability to visualize what

an action looks like from different views, because human brains can build view-invariant action representations immediately [18]. We hypothesize that enabling a deep network with the ability to extrapolate action across different views can encourage it to learn view-invariant representations. In this work, we propose an unsupervised learning framework, where the task is to construct the 3D motions for multiple target views using the video representation from a source view. We argue that in order for the network to infer cross-view motion dynamics, the learned representations should reside in a view-invariant discriminative space for action recognition.

View-invariant representation learning for cross-view action recognition has been widely studied [21, 28, 44, 45, 59, 64]. However, most of the existing methods require access to 3D human pose information during training, while others compromise discriminative power to achieve view invariance. We focus on inferring motions rather than tracking body keypoints over space and time. Our method learns a recurrent encoder which extracts motion dynamics insensitive to viewpoint changes. We represent motion as 3D flow calculated using RGB-D data only.

The contributions of our work are as follows:

- We propose an unsupervised framework to effectively learn view-invariant video representation that can predict motion sequences for multiple views. The learned representation is extracted from a CNN+RNN based encoder, and decoded into multiple sequences of 3D flows by CNN decoders. The framework is trained by jointly minimizing several losses.

- We propose a view-adversarial training to encourage view-invariant feature learning. Videos from different views are mapped to a shared subspace where a view classifier cannot discriminate them. The shared representation is enforced to contain meaningful motion information by the use of flow decoders.

- We demonstrate the effectiveness of our learned representation on cross-subject and cross-view action recognition tasks. We experiment with various input modalities including RGB, depth and flow. Our method outperforms state-of-the-art unsupervised methods across multiple datasets.

## 2 Related Work

### 2.1 Unsupervised Representation Learning

While deep networks have shown dominant performance in various computer vision tasks, the fully supervised training paradigm requires vast amount of human-labeled data. The inherent limitation highlights the importance of unsupervised learning, which leverages unlabeled data to learn feature representations. Over the past years, unsupervised learning methods have been extensively studied for deep learning methods, such as Deep Boltzmann Machines [47] and auto-encoders [1, 2, 22, 55]. Unsupervised representation learning has proven to be useful for several supervised tasks, such as pedestrian detection, object detection and image classification [5, 11, 36, 48, 61].

In the video domain, there are two lines of recent works on unsupervised representation learning. The first line of works exploit the temporal structure of videos to learn visual representation with sequence verification or sequence sorting task [8, 23, 34]. The second line of works are based on frame reconstruction. Ranzato *et al*. [46] proposed a RNN model to predict missing frames or future frames from an input video sequence. Srivastava *et al*. [53] extended this framework with LSTM encoder-decoder model to reconstruct input sequence and predict future sequence. While the above representation learning mostly capture semantic features, Luo *et al*. [33] proposed an unsupervised learning framework that predicts future 3D motions from a pair of consecutive frames. Their learned representations show promising results for supervised action recognition. However, previous works often learn view-specific representations which are sensitive to viewpoint changes.

### 2.2 Action Recognition

**RGB Action Recognition.** Action recognition from RGB videos is a long-standing problem. A detailed survey can be found in [4]. Recent approaches have shown great progress in this field, which can be generally divided into two categories. The first category focuses on designing handcrafted features for video representation, where the most successful example is improved dense trajectory features [56] combined with Fisher vector encoding [38]. The second category uses deep networks to jointly learn feature representation and classifier. Simonyan and Zisserman [51] proposed two-stream

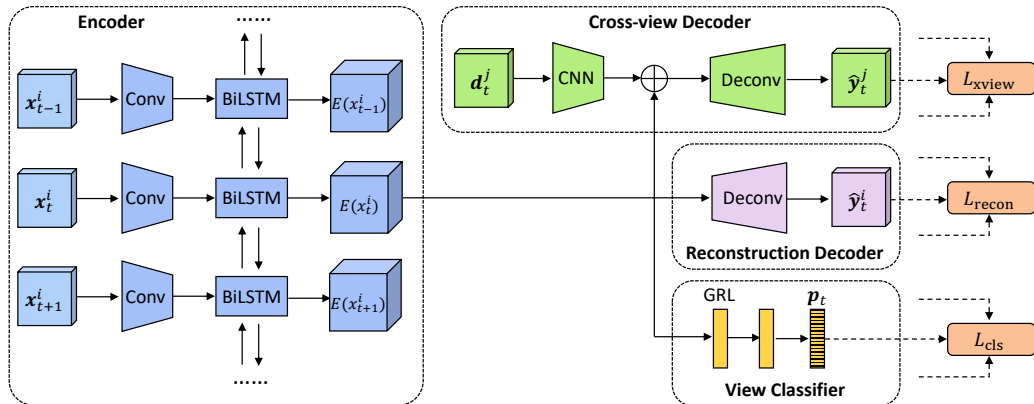

Figure 1: The proposed unsupervised representation learning framework. For a sequence of input frames, the encoder generates a sequence of feature representations. At each timestep, the representation is used by the cross-view decoder, reconstruction decoder and view classifier, where multiple loss terms are jointly minimized. The encoder can learn to generate view-invariant representations that capture motion dynamics.

CNNs, which extracts spatial and motion representation from video frames and optical flows. RNN based architectures have also been proposed to model the temporal information [6, 35]. However, deep networks training requires large amount of human-labeled data. CNNs pre-trained with ImageNet are commonly adopted as backbone [3, 6, 35, 51, 27, 26], to facilitate training and avoid overfitting.

**RGB-D Action Recognition.** Since the first work on action recognition using depth maps [29], researchers have proposed methods for action recognition that extract features from multi-modal data, including depth, RGB, and skeleton [7, 13, 16, 30, 31, 32, 39, 44, 45, 50, 57, 59]. Recently, Wang *et al.* [60] used 3D scene flow [19] calculated with RGB-D data as input for action recognition. State-of-the-art methods for RGB-D action recognition report human level performance on well-established datasets such as MSR-DailyActivity3D [57].However, [49] shows that there is a big performance gap between human and existing methods on the more challenging NTU-RGBD dataset [49], which contains significantly more subjects, viewpoints, action classes and backgrounds.

**View-invariant Feature Representation.** One particularly challenging aspect of action recognition is to recognize actions from varied unknown and unseen views, referred to as *cross-view* action recognition in the literature. The performance of most existing methods [6, 32, 35, 38, 39, 51, 56] drop sharply as the viewpoint changes due to the inherent view dependence of the features used by these methods. To tackle this problem, researchers have proposed methods to learn representations invariant to viewpoint changes. Some methods create spatial-temporal representations that are insensitive to view variations [25, 40, 59], while other methods find a view independent latent space in which features extracted from different views are directly comparable [28, 45, 64]. For example, Rahmani *et al.* [45] used a deep network to project dense trajectory features from different views into a canonical view. However, most of the previous methods require access to 3D human pose information (*e.g.* mocap data [45], skeleton [59]) during training, while others are limited by their discriminative power. Moreover, existing methods other than skeleton based methods [30, 31, 24, 42, 63] have not shown effective performance on the cross-view evaluation for NTU-RGBD dataset.

## 3 Method

The goal of our unsupervised learning method is to learn video representations that capture view-invariant motion dynamics. We achieve this by training a model that uses the representation to predict sequences of motion for multiple views. The motions are represented as 3D dense scene flows, calculated using the primal-dual method [19] with RGB-D data. The learned representation can then be used as a discriminative motion feature for action recognition. In this section, we first present the unsupervised learning framework, followed by the action recognition method.

## 3.1 Learning Framework

An overview of the learning framework is illustrated in Figure 1. It is an end-to-end deep network that consists of four components: *encoder*, *cross-view decoder*, *reconstruction decoder*, and *view classifier*, parameterized by $\{\boldsymbol{\theta}_e, \boldsymbol{\theta}_d, \boldsymbol{\theta}_r, \boldsymbol{\theta}_g\}$ respectively. The goal is to minimize the following loss:

$$\mathcal{L} = \mathcal{L}_{\mathrm{xview}} + \alpha \mathcal{L}_{\mathrm{recon}} + \beta \mathcal{L}_{\mathrm{cls}}, \tag{1}$$

where $\alpha$ and $\beta$ are weights to balance the interaction of the loss terms. $\mathcal{L}_{\mathrm{xview}}$ is the cross-view flow prediction loss, $\mathcal{L}_{\mathrm{recon}}$ is the flow reconstruction loss, and $\mathcal{L}_{\mathrm{cls}}$ is the view classification loss applied in an adversarial setting to enhance view invariance. Each loss term involves the *encoder* and one other component. Next we explain each component in detail.

**Encoder.** Let $V$ denote all the available views of an captured action. The encoder, parameterized by $\boldsymbol{\theta}_e$, takes as input a sequence of frames for view $i \in V$, denoted as $\boldsymbol{X}^i = \{\boldsymbol{x}_1^i, \boldsymbol{x}_2^i...\boldsymbol{x}_T^i\}$, and encodes them into a sequence of low-dimensionality feature embeddings $E(\boldsymbol{X}^i; \boldsymbol{\theta}_e) = \{E(\boldsymbol{x}_1^i), E(\boldsymbol{x}_2^i)...E(\boldsymbol{x}_T^i)\}$. Specifically, for each frame, we first use a downsampling CNN (denoted as "Conv") to extract a low-dimensionality feature of size $h \times w \times k$. Then, a bi-directional convolutional LSTM (denoted as "BiLSTM") runs through the sequence of extracted Conv features. At each timestep $t$, the BiLSTM generates two feature maps of size $h \times w \times k$, one through forward pass and the other through backward pass. The two feature maps are concatenated channel-wise to form the encoding $E(\boldsymbol{x}_t^i)$ of size $h \times w \times 2k$. In this work, we set $h = w = 7$ and $k = 64$.

Compared with vanilla LSTMs, convolutional LSTMs [41] replace the fully connected transformations with spatial convolutions, which can preserve spatial information in intermediate representations. We find it to perform much better than vanilla LSTMs. Moreover, our bi-directional LSTM aggregates information from both previous frames and future frames, which helps to generate richer representations. Compared with the LSTM encoder in [33] that only encodes 2 frames, the proposed encoder can generate encodings for longer sequences, which embodies more discriminative motion dynamics for the action. In this work, we set the sequence length $T = 6$.

**Cross-view decoder.** The goal of the cross-view decoder is to predict the 3D flow $\boldsymbol{y}_t^j$ for view $j$ ($j \in V; j \neq i$), given the encoding $E(\boldsymbol{x}_t^i)$ for view $i$, at timestep $t$. Inferring $\boldsymbol{y}_t^j$ directly from $E(\boldsymbol{x}_t^i)$ is too difficult, because the decoder has zero information about view $j$. Therefore, we give an additional input to the decoder that contains view-specific information. This input is the depth map $\boldsymbol{d}_t^j$ for view $j$ at timestep $t$, which serves as an anchor to inform the decoder about the spatial configuration of view $j$. The decoder still requires view-invariant motion dynamics from $E(\boldsymbol{x}_t^i)$ in order to predict $\boldsymbol{y}_t^j$.

Specifically, we first use a CNN to extract a feature of size $h \times w \times k$ from $\boldsymbol{d}_t^j$. The extracted feature is concatenated with $E(\boldsymbol{x}_t^i)$ channel-wise into a feature of size $h \times w \times 3k$. Then we use an upsampling CNN (denoted as "Deconv") to perform spatial upsampling. Deconv consists of four fractionally-strided convolutional layers [52] with batch normalization layer [17] and ReLU activation in between. We observe that the batch normalization is critical to optimize the network.

Let $\hat{\boldsymbol{y}}_t^j = D(E(\boldsymbol{x}_t^i), \boldsymbol{d}_t^j; \boldsymbol{\theta}_d)$ denote the output of the cross-view decoder for timestep $t$. We want to minimize the mean squared error between $\hat{\boldsymbol{y}}_t^j$ and $\boldsymbol{y}_t^j$ for $t = 1, 2...T$:

$$\mathcal{L}_{\mathrm{xview}}^j(E(\boldsymbol{X}^i), \boldsymbol{D}^j, \boldsymbol{Y}^j) = \sum_{t=1}^{T} \left\| \boldsymbol{y}_t^j - \hat{\boldsymbol{y}}_t^j \right\|_2^2, \tag{2}$$

where $\boldsymbol{D}^j = \{\boldsymbol{d}_1^j, \boldsymbol{d}_2^j...\boldsymbol{d}_T^j\}$ is the sequence of anchor depth frames, and $\boldsymbol{Y}^j = \{\boldsymbol{y}_1^j, \boldsymbol{y}_2^j...\boldsymbol{y}_T^j\}$ is the sequence of flows.

Since we want to learn a video representation that can be used to predict motions for multiple views, we deploy multiple cross-view decoders with shared parameters to all views other then $i$. Therefore, the cross-view flow prediction loss for $\boldsymbol{X}^i$ is:

$$\mathcal{L}_{\mathrm{xview}}(\boldsymbol{X}^i) = \sum_{j} \mathcal{L}_{\mathrm{xview}}^j(E(\boldsymbol{X}^i), \boldsymbol{D}^j, \boldsymbol{Y}^j) \qquad \text{for } j \in V; j \neq i \tag{3}$$

**Reconstruction decoder.** The goal of the this decoder is to reconstruct the 3D flow $\boldsymbol{y}_t^i$ given the encoding for the same view $E(\boldsymbol{x}_t^i)$. Learning flow reconstruction helps the encoder to extract basic motions, and when used together with cross-view decoders, enhances learning of view-invariant

motion dynamics. The architecture of the reconstruction decoder is a Deconv module similar as cross-view decoder, with the number of input channels in the first layer adapted to $2k$. Let $\hat{\boldsymbol{y}}_t^i = R(E(\boldsymbol{x}_t^i); \boldsymbol{\theta}_r)$ be the output of the reconstruction decoder at timestep $t$, the flow reconstruction loss is:

$$\mathcal{L}_{\text{recon}}(\boldsymbol{X}^i, \boldsymbol{Y}^i) = \sum_{t=1}^{T} \left\| \boldsymbol{y}_t^i - \hat{\boldsymbol{y}}_t^i \right\|_2^2 \tag{4}$$

**View classifier.** We propose a view-adversarial training that encourages the encoder to learn video representations invariant to view changes. We draw inspiration from the domain-adversarial training [9, 10], which aims at learning features that are indiscriminate with respect to shift between domains. The proposed view-adversarial training is achieved by adding a view classifier connected to the encoder through a Gradient Reversal Layer (GRL). The view classifier tries to predict which view the encoded representation belongs to, whereas the encoder tries to confuse the view classifier by generating view-invariant representations.

More formally, the view classifier $G(E(\boldsymbol{x}_t^i); \boldsymbol{\theta}_g) \to \boldsymbol{p}_t$ maps an encoding at timestep $t$ to a probability distribution over possible views $V$. Learning with GRL is adversarial in that $\boldsymbol{\theta}_g$ is optimized to increase $G$'s ability to discriminate encodings from different views, while GRL reverses the sign of the gradient that flows back to $E$, which results in the encoder parameters $\boldsymbol{\theta}_e$ learning representations that reduces the view classification accuracy. Essentially, we *minimize* the cross-entropy loss for the view classification task with respect to $\boldsymbol{\theta}_g$, while *maximize* it with respect to $\boldsymbol{\theta}_e$. Therefore, we define the view classification loss as the sum of the cross-entropy loss for the entire sequence:

$$\mathcal{L}_{\text{cls}}(\boldsymbol{X}^i) = \sum_{t=1}^{T} -\log\big(p_t^i\big), \tag{5}$$

where $i$ is the ground-truth view of the input.

The view classifier consists of two fully connected layers and a softmax layer. Since the encoding $E(\boldsymbol{x}_t^i)$ is a convolutional feature, it is first flattened into a vector before it goes into the view classifier.

### 3.2 Action Recognition

We use the encoder from unsupervised learning for action recognition. Given the learned representations for a sequence of frames $E(\boldsymbol{X}) = \{E(\boldsymbol{x}_t)|t = 1, 2...T\}$, we apply an action classifier to each $E(\boldsymbol{x}_t)$. The action classifier is a simple fully-connected layer, which takes the flattened vector of $E(\boldsymbol{x}_t)$ as input, and outputs a score $\boldsymbol{s}_t$ over possible action classes. The final score of the sequence is the average score for each timestep: $\boldsymbol{s} = \frac{1}{T}\sum_{t=1}^{T} \boldsymbol{s}_t$.

The action classifier is trained with cross-entropy loss. During training, we consider three scenarios:

(a) *scratch*: Randomly initialize the weights of encoder and train the entire model from scratch.

(b) *fine-tune*: Initialize the encoder with learned weights and fine-tune it for action recognition.

(c) *fix*: Keep the pre-trained encoder fixed and only train the action classifier.

At test time, we uniformly sample 10 sequences from each video with sequence length $T = 6$, and average the scores across the sampled sequences to get the class score of the video.

## 4 Experiments

### 4.1 Unsupervised Representation Learning

**Implementation details.** For Conv in encoder and depth CNN in cross-view decoder, we employ the ResNet-18 architecture [15] up until the final convolution layer, and add a $1 \times 1 \times 64$ convolutional layer to reduce the feature size. The number of input channels in the first convolutional layer is adapted according to input modality. Note that our CNN has not been pre-trained on ImageNet. For BiLSTM, we use convolutional filters of size $7 \times 7 \times 64$ for convolution with input and hidden state. We initialize all weights following the method in [14]. During training, we use a mini-batch of size 8. We train the model using the Adam optimizer [20], with an initial learning rate of $1e^{-5}$ and a weight decay of $5e^{-4}$. We decrease the learning rate by half every 20000 steps (mini-batches). To avoid

Table 1: Cross-view flow prediction error on NTU RGB+D dataset.

| Method | Cross-subject | | | Cross-view | | |
|---|---|---|---|---|---|---|
| | RGB | Depth | Flow | RGB | Depth | Flow |
| proposed method w/o $\mathcal{L}_{\text{recon}}$ & $\mathcal{L}_{\text{cls}}$ | 0.0267 | 0.0244 | 0.0201 | 0.0265 | 0.0238 | 0.0199 |
| proposed method w/o $\mathcal{L}_{\text{cls}}$ | 0.0259 | 0.0235 | 0.0198 | 0.0252 | 0.0223 | 0.0194 |
| proposed method | 0.0254 | 0.0229 | 0.0193 | 0.0248 | 0.0220 | 0.0193 |

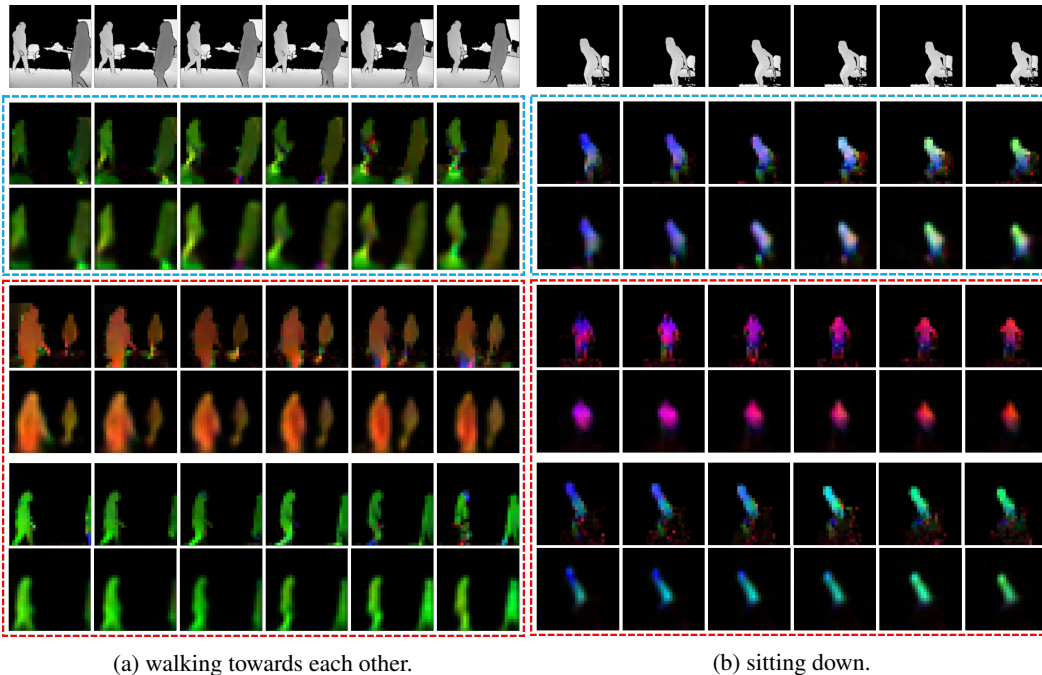

(a) walking towards each other.      (b) sitting down.

Figure 2: Example of flow sequences with depth input. 3D flows are visualized as RGB images. Upper rows are ground-truth flows, and lower rows are predicted flows. Blue box denotes source view flow reconstruction, whereas red box denotes cross-view flow prediction. The model can estimate raw motions for multiple views.

distracting the flow prediction task, we activate the view adversarial training after 5000 steps. The weights of the loss terms are set as $\alpha = 0.5$ and $\beta = 0.05$, which is determined via cross-validation.

In order to effectively predict the motions, we want to describe the motion as low-dimensional signal. Hence, we apply spatial downsampling to the 3D flows by calculating the mean of each non-overlapping $8 \times 8$ patch. The resulting $28 \times 28 \times 3$ flow maps are multiplied by 50 to keep a proper scale, which become the ground-truth $\boldsymbol{Y}$.

**Dataset.** We use the NTU RGB+D dataset [49] for unsupervised representation learning. The dataset consists of 57K videos for 60 action classes, captured from 40 subjects in 80 camera viewpoints. The 80 viewpoints can be divided into five main views based on the horizontal angle of the camera with respect to the subject: front view, left side view, right side view, left side 45 degrees view and right side 45 degrees view. These five views form the view set used in our experiments. Each action sequence is simultaneously captured by three cameras from three of the five views at a time.

**Evaluation.** There are two sets of standard evaluation protocols for action recognition on NTU RGB+D dataset: cross-subject evaluation and cross-view evaluation. Following it, we conduct two unsupervised learning experiments, where in each experiment we ensure that the encoder will not be trained on any test samples in supervised learning setting. For cross-subject evaluation, we follow the same training and testing split as in [49]. For cross-view evaluation, samples of cameras 2 and 3 are used for training while those of camera 1 for testing. Since we need at least two cameras for our unsupervised task, we randomly divide the supervised training set with ratio of 8:1 for unsupervised training and test. We use the cross-view flow prediction loss $\mathcal{L}_{\text{xview}}$ as the evaluation metric, which

Table 2: Action recognition accuracy (%) on NTU RGB+D dataset.

| Method | Cross-subject | | | Cross-view | | |
|---|---|---|---|---|---|---|
| | RGB | Depth | Flow | RGB | Depth | Flow |
| scratch | 36.6 | 42.3 | 70.2 | 29.2 | 37.7 | 72.6 |
| fix | 48.9 | 60.8 | 77.0 | 40.7 | 53.9 | 78.8 |
| fine-tune w/o $\mathcal{L}_{\mathrm{recon}}$ & view-adversarial | 51.1 | 63.5 | 79.7 | 42.9 | 57.0 | 79.9 |
| fine-tune w/o view-adversarial | 53.4 | 66.0 | 80.3 | 46.2 | 60.1 | 81.9 |
| fine-tune | 55.5 | 68.1 | **80.9** | 49.3 | 63.9 | **83.4** |

Table 3: Comparison with state-of-the-art methods for action recognition on NTU RGB+D dataset.

| Method | Modality | Cross-subject | Cross-view |
|---|---|---|---|
| HOG [37] | | 32.24 | 22.27 |
| Super Normal Vector [62] | | 31.82 | 13.61 |
| HON4D [39] | Depth | 30.56 | 7.26 |
| *Shuffle and Learn* [34] | | 46.2 | 40.9 |
| *Luo et al.* [33] | | 61.4 | 53.2 |
| Ours | | **68.1** | **63.9** |
| Lie Group [54] | | 50.08 | 52.76 |
| FTP Dynamic Skeletons [16] | | 60.23 | 65.22 |
| HBRNN-L [7] | | 59.07 | 63.97 |
| 2 Layer P-LSTM [49] | | 62.93 | 70.27 |
| ST-LSTM [30] | Skeleton | 69.2 | 77.7 |
| GCA-LSTM [31] | | 74.4 | 82.8 |
| Ensemble TS-LSTM [24] | | 74.60 | 81.25 |
| Depth+Skeleton [42] | | 75.2 | 83.1 |
| VA-LSTM [63] | | 79.4 | **87.6** |
| Ours | Flow | **80.9** | **83.4** |

quantifies the performance of the model to predict motions across different views. We experiment with three input modalities: RGB, depth and 3D flow.

**Results.** Table 1 shows the quantitative results for the unsupervised flow prediction task. In order to demonstrate the effect of different components and loss terms, we evaluate different variants of the proposed framework. First, we only train the encoder with cross-view decoder (denoted as proposed method w/o $\mathcal{L}_{\mathrm{recon}}$&$\mathcal{L}_{\mathrm{cls}}$). Then we add the reconstruction decoder with flow reconstruction loss (denoted as proposed method w/o $\mathcal{L}_{\mathrm{cls}}$). Finally we add view adversarial training with view classification loss to form the proposed method. Across all input modalities, flow reconstruction and view adversarial training both can improve the cross-view flow prediction performance. Comparing between different input modalities, flow achieves the lowest $\mathcal{L}_{\mathrm{xview}}$. This is expected because flow contains more view-invariant motion information.

Figure 2 shows qualitative examples of flow prediction with depth input. For each pair of rows, the upper rows are ground-truth flows, whereas the lower rows are flows predicted by the decoders. The model shows the ability to estimate raw motions for multiple views with the encoded representations.

## 4.2 Action Recognition on NTU RGB+D

**Implementation details.** We experiment with the three settings described in Section 3.2. We train the model using Adam optimizer [20], with the mini-batch size as 16, learning rate as $1e^{-4}$ and weight decay as $5e^{-4}$. We set the learning rate of the encoder to be $1e^{-5}$ for *fine-tune*. For *scratch*, we decay the learning rate by half every 20000 steps. For *fine-tune* and *fix*, since training converges faster, we half the learning rate every 10000 steps.

**Results.** Table 2 shows the classification accuracy for both cross-subject and cross-view action recognition with three input modalities. Across all modalities, supervised learning from scratch

Table 4: Cross-subject action recognition accuracy (%) on MSRDailyActivity3D dataset.

| Method | Accuracy |
|---|---|
| Actionlet Ensemble [58] (S) | 85.8 |
| HON4D [39] (D) | 80.0 |
| MST-AOG [59] (D) | 53.8 |
| SNV [62] (D) | 86.3 |
| HOPC [43] (D) | 88.8 |
| *Luo et al.* [33] (D) | 75.2 |
| Ours (scratch) | 42.5 |
| Ours (fine-tune) | 82.3 |

Table 5: Cross-view action recognition accuracy (%) on Northwestern-UCLA dataset.

| Method | Accuracy |
|---|---|
| Actionlet Ensemble [58] (S) | 69.9 |
| Hankelets [25] | 45.2 |
| MST-AOG [59] (D) | 53.6 |
| HOPC [43] (D) | 71.9 |
| R-NKTM [45] (S) | 78.1 |
| *Luo et al.* [33] (D) | 50.7 |
| Ours (scratch) | 35.8 |
| Ours (fine-tune) | 62.5 |

has the lowest accuracy. Using the unsupervised learned representations and training only a linear action classifier (*fix*) significantly increases accuracy. Fine-tuning the encoder can further improve performance. If we remove the view-adversarial training in the unsupervised framework, the accuracy would decrease, especially for cross-view recognition.

Among the three input modalities, flow input achieves the highest accuracy, which agrees with our unsupervised learning result. Flow is also the only input modality that has a higher accuracy for cross-view recognition compared with cross-subject recognition. This supports our observation that flow is more view-invariant than the other two modalities.

**Comparison with state-of-the-art.** In Table 3 we compare our method against state-of-the-art methods on NTU RGB+D dataset. The first group of methods use depth as input, and the second group of methods use skeleton as input. We re-implement two unsupervised learning methods [33, 34] (in *italic*) and report their classification accuracy. We do not directly cite the results in [33] because [33] reports mAP rather than accuracy. Our re-implementation achieve similar mAP as in [33].

Using depth input, the proposed method outperforms all previous methods. The increase in accuracy is more significant for cross-view recognition, which shows that the learned representation is invariant to viewpoint changes. Using flow input, our method achieves comparable performance to skeleton-based methods. However, skeleton is a higher level feature that is more robust to viewpoint change. Moreover, the method [63] with higher cross-view accuracy uses explicit coordinate system transformation to achieve view invariance.

## 4.3 Transfer Learning for Action Recognition

In this section, we perform transfer learning tasks, where we use the unsupervised learned representations for action recognition on two other datasets in new domains (different subjects, environments and viewpoints). We perform cross-subject evaluation on MSR-DailyActivity3D Dataset, and cross-view evaluation on Northwestern-UCLA MultiviewAction3D Dataset. We experiment with *scratch* and *fine-tune* settings, using depth modality as input.

**MSR-DailyActivity3D Dataset.** This dataset contains 320 videos of 16 actions performed by 10 subjects. We follow the same experimental setting as [57], using videos of half of the subjects as training data, and videos of the rest half as test data.

**Northwestern-UCLA MultiviewAction3D Dataset.** This dataset contains 1493 videos of 10 actions performed by 10 subjects, captured by 3 cameras from 3 different views. We follow [59] and use videos from the first two views for training and videos from the third view for test.

**Results.** Table 4 and 5 show our results in comparison with state-of-the-art methods. On both datasets, training a deep model from scratch gives poor performance. Using the unsupervised learned representations increases the accuracy by a large margin. Our method outperforms previous unsupervised method [33], and achieves comparable performance with skeleton-based methods (marked by S) and depth-based methods (marked by D) that use carefully hand-craft features. This demonstrates that the learned representations can generalize across domains.

## 5 Conclusion

In this work, we propose an unsupervised learning framework that leverages unlabeled video data from multiple views to learn view-invariant video representations that capture motion dynamics. We learn the video representations by using the representations for a source view to predict the 3D flows for multiple target views. We also propose a view-adversarial training to enhance view-invariance of the learned representations. We train our unsupervised framework on NTU RGB+D dataset, and demonstrate the effectiveness of the learned representations on both cross-subject and cross-view action recognition tasks across multiple datasets.

The proposed unsupervised learning framework can be naturally extended beyond actions. For future work, we intend to extend our framework for view-invariant representation learning in other tasks such as gesture recognition and person re-identification. In addition, we can consider generative adversarial network (GAN) [12] for multi-view data generation.

### Acknowledgments

This research is supported by the National Research Foundation, Prime Minister's Office, Singapore under its Strategic Capability Research Centres Funding Initiative.

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
