[Reviews · NeurIPS 2018]

Reviewer 1



The authors present an unsupervised learning framework to recognize actions in videos. They propose to learn a representation that predicts 3D motion in other viewing angles. They leverage a view-adversarial training method. They run experiments on multi-modal datasets for action recognition. Pros: - Unsupervised learning for video is a relevant problem - The suggested surrogate task that learns view-invariant representation is interesting. - It s interesting to see that their proposed method based on flow input outperforms other methods... Cons: - l25: The claim for view specific is debatable. Some of the previous works, e.g.. [31], models 3D information to be view invariant…Their performance is the same the proposed method although the authors report different number (see next bullet).
 - In table 3&4, why are the authors re-implementing [31] and [32] since the exact numbers are available in the original papers for the same used datasets? In fact, the authors are reporting numbers for other methods AND NOT for these 2 ones. Why not do the same for 31 and 32? At least, the authors can share the reported numbers as well as the ones with their own re-implemented version. Discarding numbers is not correct. Similarly for table 4, the performance for [4] is different from previous papers. Why is it 80 instead of 85? That again communicates a questionable performance evaluation. Because of the above weaknesses, I have strong doubt on the performance evaluation section. I have identified few inconsistencies. Hence, I am not confident that the proposed method outperforms previous methods.

Reviewer 2



This paper addresses the problem of view-invariant action representation within an unsupervised learning framework. In particular, the unsupervised learning task is the prediction of 3D motion from different viewpoints. The proposed model comprises four modules: "encoder", "cross-view decoder", "reconstruction decoder" and "view classifier". For training purposes, a loss function defined as the linear combination of three task-specific losses is proposed. Given an encoding, the cross-view decoder is in charge of estimating the 3D flow in a target view different of the source one. Then, the goal of the reconstruction decoder is to reconstruct the 3D flow of a target view given the encoding for the same view. And, finally, the view classifier is trained in an adversarial way, i.e., the view classifier tries to classify the given encoding of the input, while the adversarial component promotes the learning of view-invariant encodings. To deal with temporal information, a bi-directional convolutional LSTM is used. Three modalities have been considered during the experimental validation: RGB, depth and flow. The experimental results on three benchmarks demonstrate the validity of the proposed framework. Overall, in my opinion, this is a high-quality work of wide interest for progressing in the "unsupervised motion representation" problem. * Strengths: + The problem of unsupervised learning of 3D motion is of interest for the computer vision community in general, and for NIPS community in particular, as it has been already addressed in top-tier conferences and journals. + It is very interesting the combination of the idea of using cross-reconstruction of 3D flow and adversarial training for view classification from the encodings. + The paper is well-written and easy to read. + Competitive results are obtained on three standard datasets for action recognition: NTU RGB+D, MSRDailyActivity3D and Northwestern-UCLA. The experimental results support the viability of the proposal. * Weaknesses: - In my humble opinion, further implementation details would be needed to be able to fully reproduce the presented results. * Detailed comments: - Sec. 3.1, in Encoder, how has the value of $T=6$ been selected? How does this value influence in the performance of the system? - In Tab. 3, column 'Cross-subject', row 'Skeleton', I guess number 79.4 should be highlighted in bold. * Minor details: - Page 3, line 89: extra space is needed after "...[55]. " - Page 4, line 152, "...other then i..." --> "than" [About author feedback] I have carefully read the author response and my positive "overall score" remains the same.

Reviewer 3



The paper presents a method for learning view-invariant action representations for flow. The method consists of an encoder (convolutional and bi-directional convolutional LSTMs), which learns feature representations for various views up to T=6 frames, and three additional components: (a) a cross-view decoder, which learns to predict motions for multiple views (different than the ones of the encoder), (b) a reconstruction decoder, which reconstructs the input flow, and (c) a view classifier, which predicts from which view the encoded representation comes and operates under an adversarial setting, where the encoder tries to generate view-invariant representations while the view classifier tries to distinguish their view. The paper is well-written and easy to follow. It tackles the difficult problem of unsupervised learning of view-invariant representations for flow. The choice of the architecture is interesting. Nevertheless, the reasons for using the view classifier branch are not clear. Given that the goal is to learn view-invariant representations, the view classifier operates against this goal. This is also in line with the small value of the weight of the decoder beta (line 200 for equation 1), which is 0.05. This shows that the view classifier barely contributes to the final loss. Moreover, the ablation study of Table 1 shows that this branch barely changes the performance (flow input/output for cross view). It would be interesting to know if the authors have an explanation for this choice. The method seems clean but complicated to implement and run. It would be interesting to see some run times, especially to examine the overload of the view classifier. The ablation study of Table 1 is performed using the Lxview. It would be beneficial to see the total loss of equation 1 (with and w/o some components). Moreover, it would be interesting to see how the evolution of the classification accuracy also by removing the reconstruction decoder in Table 2. In the whole experimental section the authors use one modality (RGB, depth or flow) as input and output. It would be interesting to see if the combination of the modalities could also help, at least as an ablation study (Table 1) or even in Tables 2 and 3. In the ablation study of Table 1, the authors evaluate the loss with RGB, depth and flow inputs for flow outputs. The cross-view decoder, however, takes as additional input a depth map for a view. Therefore, the columns of the table should display that the input is not one single modality but it includes depth in all cases. The same holds for Tables 2 and 3. For the cross-view decoder, in line 152, the authors state that they deploy multiple view decoders. Is the number of these decoders fixed, eg. 5 (line 207) or do the authors allow the network to learn various views even if they do not exist at test time? For action recognition, training from scratch (encoder + action classifier) performs significantly worse than all other cases, where the encoder is learnt using the proposed unsupervised setting. I believe that here another baseline is missing. The authors could have used pre-trained models on ImageNet and/or UCF-101 and/or Kinetics to initialize the encoder and then fine-tune it. It would have been a stronger baseline. In the action recognition Section 3.2, the architecture of the action classifier does not seem complete. The authors use only one fully-connected layer for as a classifier. It would be interesting to understand this choice and also to see some results with something more complete, eg. at least another one fully-connected layer. It would be great if the authors could explain how they ended up in the values for the weights a=0.5 and beta=0.05 (lines 200) for equation (1). Did they use cross-validation? The results in Figure 2 look nice. It would be interesting to see also some failure cases and some discussion for them. Some choices of architectural parameters are not really justified. For instance, it is not clear how the authors chose k=64 (line 128), T=6 (line 135). In Table 4, for the Northwestern-UCLA Multiview Action 3D dataset recent works typically report results in all split combinations (training on two splits and evaluating on the third one) and also include their average. In Table 3, for NTU RGB+D the average is also missing. The authors have missed some state of the art citations in Tables 3, 4, and 5. For instance: [42], - Liu, Mengyuan, Hong Liu, and Chen Chen. "Enhanced skeleton visualization for view invariant human action recognition." In Pattern Recognition 2017 - Baradel, Fabien, Christian Wolf, and Julien Mille. "Human action recognition: Pose-based attention draws focus to hands." In ICCV Workshop on Hands in Action 2017. - Baradel, Fabien, Christian Wolf, Julien Mille, and Graham W. Taylor. "Glimpse clouds: Human activity recognition from unstructured feature points." In CVPR 2018. - Rahmani, Hossein, and Ajmal Mian. "Learning a non-linear knowledge transfer model for cross-view action recognition." In CVPR 2015. - Gupta, Ankur, Julieta Martinez, James J. Little, and Robert J. Woodham. "3D pose from motion for cross-view action recognition via non-linear circulant temporal encoding." In CVPR 2014. --after feedback The authors addressed many of the issues raised in the reviews. I am still not convinced about the Lcls: even though its value is sometimes higher than the other losses (without any explanation), using it does not seem so crucial (Table 1 and Table 2). However, the idea of casting the problem as a view-adversarial training is indeed interesting. Including results with pre-training on ImgNet and potentially on other datasets (eg. Kinetics) is important and the rebuttal shows that pre-training on ImgNet is inferior to fix. The authors should also include all related works and results.